

# A combination of two ELISA tests for nasopharyngeal carcinoma screening in endemic areas based on a case-control study

Dongping Rao[1,*], Meiqin Fu[2,*], Yingjie Chen[2], Qing Liu[3], Lin Xiao[4], Xin Zhang[5], Zhongxiao Li[2], Haitao Li[2], Yongyi He[1], Yongxing Chen[6], Jieying Chen[2], Jin Hu[2] and Yanming Huang[7]

[1] Department of Medical Records, Jiangmen Central Hospital, Affiliated Jiangmen Hospital of Sun Yat-sen University, Jiangmen, Guangdong, China

[2] Clinical Laboratory, Jiangmen Central Hospital, Affiliated Jiangmen Hospital of Sun Yat-sen University, Jiangmen, Guangdong, China

[3] Department of Preventive Medicine, Sun Yat-sen University Cancer Center, Guangzhou, Guangdong, China

[4] Department of Radiotherapy, Jiangmen Central Hospital, Affiliated Jiangmen Hospital of Sun Yat-sen University, Jiangmen, Guangdong, China

[5] Clinical Experimental Center, Jiangmen Key Laboratory of Clinical Biobanks and Translational Research, Jiangmen Central Hospital, Affiliated Jiangmen Hospital of Sun Yat-sen University, Jiangmen, Guangdong, China

[6] Department of Ear-Nose-Throat, Jiangmen Central Hospital, Affiliated Jiangmen Hospital of Sun Yat-sen University, Jiangmen, Guangdong, China

[7] Department of Respiratory Medicine, Clinical Experimental Center, Jiangmen Key Laboratory of Clinical Biobanks and Translational Research, Jiangmen Central Hospital, Affiliated Jiangmen Hospital of Sun Yat-sen University, Jiangmen, Guangdong, China

* These authors contributed equally to this work.

Corresponding authors
Dongping Rao, a113202660@126.com
Yanming Huang, huangyanming_jxy@163.com

## ABSTRACT

For populations with a high risk of nasopharyngeal carcinoma (NPC) in Guangdong province in southern China, mass screening is the first choice to prevent death from NPC. To improve the performance of NPC screening, we used a combination based on the IgA antibody against the Epstein-Barr virus (EBV) capsid antigen (VCA-IgA) and the IgA antibody against Epstein-Barr virus nuclear antigen 1 (EBNA1-IgA) to NPC screening by enzyme-linked immunosorbent assay (ELISA). A multiplication model was applied to measure the level of the combination. We evaluated the NPC screening effect of the markers.A case-control study was performed to assess the NPC screening effect of the markers. A total of 10,894 serum specimens were collected, including 554 samples from NPC patients and 10,340 samples from healthy controls. In the training stage, 640 subjects were randomly selected, including 320 NPC cases and 320 healthy controls. In the verification stage, 10,254 subjects were used to verify the NPC screening effect of the combination. Receiver operating characteristic (ROC) analysis was performed. In the verification stage, the combination achieved an sensitivity of 91.45%, a specificity of 93.45%, and an area under the ROC curve (AUC) of 0.978 (95% CI [0.968–0.987]). Compared with VCA-IgA and EBNA1-IgA individually, the combination had an improved screening performance. A probability (PROB) calculated by logistic regression model based on VCA-IgA and EBNA1-IgA

was applied to NPC screening by ELISA in China. The AUC of the combination was a little bit larger than the PROB. There was a slight increase (3.13%) in the sensitivity of the combination compared to the sensitivity of the PROB, while the specificity was lower for the combination (92.50%) than for the PROB (95.94%). We successfully applied a combination of two ELISA tests based on VCA-IgA and EBNA1-IgA for NPC screening by using a multiplication model. The results suggested that the combination was effective and can be an option for NPC screening.

## INTRODUCTION

In southern China and southeast Asia, nasopharyngeal carcinoma (NPC) is a common malignant tumour, with an incidence rate of 10–40/100,000 per year (*Ng et al., 2005*; *Torre et al., 2015*; *Wei & Sham, 2005*; *Yang et al., 2005*; *Yu & Yuan, 2002*). Jiangmen city, an endemic area of NPC located along the Zhujiang River in the central southern area of Guangdong province, has a high-risk NPC population. The NPC incidence rate in the Jiangmen urban area is $14.99/10^5$ (*Wei et al., 2017*). The population-based cancer registry was established in Jiangmen to report the incidence and mortality of cancers. Population-based NPC screening was performed in the Jiangmen urban area by Jiangmen Central Hospital from June 2018 to March 2020.

The occurrence of NPC is strongly associated with Epstein-Barr virus (EBV) infection (*Fachiroh et al., 2004*; *Gulley, 2001*; *Henle & Henle, 1976*; *Sam, Abu-Samah & Prasad, 1994*). Furthermore, host genetics, smoking, the consumption of salted fish and occupational exposures are contributors to the pathogenesis of NPC (*Chang & Adami, 2006*; *Chang et al., 2017*; *Chen et al., 2019*; *Yong et al., 2017*). The development mechanisms of NPC are unclear. Mass screening is the main effective measure to detect NPC early in endemic areas.

EBV antibodies are widely used as markers in NPC screening (*Chien et al., 2001*; *Ji et al., 2019*; *Ji et al., 2007*; *Ng et al., 2005*; *Tan et al., 2020*; *Zeng et al., 1982*). A number of studies have shown that screening for NPC by using EBV antibodies is an effective measure to improve the survival rate of NPC patients (*Choi et al., 2011*; *Ji et al., 2007*; *Jia et al., 2006*; *Ng et al., 2010*). The combined serological test based on the IgA antibody against the EBV capsid antigen (VCA-IgA) and the IgA antibody against EBV nuclear antigen 1 (EBNA1-IgA) by enzyme-linked immunosorbent assay (ELISA) was used for NPC screening in endemic areas in China (*Gao et al., 2017*; *Liu et al., 2012*; *Yu et al., 2018*). In previous studies, the probability (PROB) calculated by logistic regression based on VCA-IgA and EBNA1-IgA was applied to NPC screening in China (*Gao et al., 2017*; *Liu et al., 2012*; *Yu et al., 2018*).

Multiplication model was applied to make new maker to improve diagnostic effect (*Enyedi et al., 2020*). In this study, a combination of two ELISA tests based on VCA-IgA and
EBNA1-IgA was applied to improve the effect of NPC screening by using a multiplication model and the NPC screening effect of the markers was evaluated.

## MATERIALS & METHODS

### Study population

A case-control study was performed to compare the effect of the NPC screening of markers, including 554 NPC cases and 10,340 healthy controls. This study included the training stage and the verification stage. The inclusion criteria for NPC cases included being histologically confirmed by biopsy, aged between 30 and 69 years, and residing in Jiangmen. A total of 554 serum specimens were continuously collected from NPC patients at Jiangmen Central Hospital from June 2018 to March 2020. Among the 554 cases, 7 (1.26%) participated in the NPC screening program. NPC stages were classified according to the 2008 staging system of China (*Lin et al., 2009*). The stages were divided into early-stage (stage I and stage II) and advanced-stage (stage III and stage IV) disease. A total of 554 cases comprised 73 early-stage cases and 481 advanced-stage cases. A total of 320 NPC training samples were randomly selected from the 554 NPC cases, and the remaining 234 of 554 NPC cases were used as validation samples.

A total of 10,340 healthy controls were obtained from an NPC screening programme performed in a population aged 30–69 years in the Jiangmen City urban area from June 2018 to March 2020. The healthy controls resided in Jiangmen of Guangdong province. A total of 320 training samples were randomly selected from the 10,340 healthy controls and were frequency matched to the 320 training NPC cases by age (5-year age groups) and sex. The remaining 10.020 of 10.340 healthy controls were used as the validation samples.

The information on age, sex, smoking history and family history of NPC for the cases and healthy controls were collected by inquiring medical records and using a questionnaire survey.

### Serological test

In total, 10,894 serum samples were collected and underwent serological tests in separate batches at Jiangmen Central Hospital. The samples were separated and stored at −40 °C. In this study, the NPC screening markers included VCA-IgA, EBNA1/IgA and combination. The antibodies VCA-IgA (Euroimmun, Lubeck, Germany) and EBNA1-IgA (Zhongshan Bio-tech, Zhongshan, China) were tested by ELISA on a TECAN Freedom EVOlyzer 200/8 platform according to the manufacturer's specifications. EBNA1s in Zhongshan Bio-tech kit were produced with purified recombinant peptide specified by EBV BKRF1 (72 kD) (*He et al., 2018*). The EBV VCAs in Euroimmun kit were obtained from the pyrolysis products of human B lymphocytes (P3HR1cell line) infected by EBV (*Gao et al., 2017*). The levels of the antibodies were assessed by the relative optical density (rOD) calculated according to the manufacturers' instructions by dividing the optical density (OD) value by a reference control (*Ji et al., 2014*). In this study, the multiplication model based on VCA-IgA and EBNA1-IgA was calculated by using the following formula: *The level of combination VCA-IgA × EBNA-IgA*. The formula for PROB calculated by logistic regression based on VCA-IgA and EBNA1-IgA was as follows:

$LogitPROB = -3.934 + 2.203 \times VCA-IgA \times EBNA-IgA$ (*Gao et al., 2017*; *Yu et al., 2018*). In the formulas, VCA-IgA and EBNA1-IgA represent the rOD values for VCA-IgA and EBNA1-IgA, respectively, which were tested by ELISA.

The written informed consent was obtained from healthy controls. The serum samples of NPC patients were collected after clinical use which were exempted from informed consent. This study was approved by the Clinical Research Ethics Committee of the Jiangmen Central Hospital (2019–28).

## Statistical analysis

Categorical variables are described as numbers and percentages. Continuous variables are shown as the means and standard deviations (SDs). The levels of VCA-IgA, EBNA1-IgA, PROB and combination were compared by $t$ tests in different population. Receiver operating characteristic (ROC) curve analysis was performed. The cut-off value of each marker was defined with the largest Youden Index selected from each ROC curve. The effects of the screening markers were measured using the sensitivity, specificity and area under the ROC curve (AUC). The base information of different populations was described and compared by the $\chi^2$ test and Fisher's exact test. The difference in sensitivities of markers were compared by $\chi^2$ test, Fisher's exact test and McNemar test.

The differences in AUCs were compared using the Z test according to the DeLong method (*DeLong, DeLong & Clarke-Pearson, 1988*). The 95% confidence intervals (CIs) of the sensitivities, specificities and AUCs were calculated. The statistical analyses were carried out using MedCalc Statistical Software version 15.2.2 (MedCalc Software bvba, Ostend, Belgium) and GraphPad Prism software version 8.0 (San Diego, CA, USA) and were two-sided, with significance set at $p < 0.05$.

# RESULTS

## Baseline information

The characteristics of the 554 cases and 10340 healthy controls are shown in Table 1. In total, 554 NPC patients were enrolled in this study. Of them, 397 (71.66%) were men, and the mean age was $50.86 \pm 9.48$ years. Among the 554 patients, 198 (35.74%) had a smoking history, and 56 (10.11%) had a family history of NPC. Of the 10340 healthy controls, 3959 (38.29%) were men, and the mean age was $48.57 \pm 11.60$ years. Among the 10,340 healthy controls, 1,670 (16.15%) had a smoking history, and 198 (1.91%) had a family history of NPC (Table 1).

The differences in age, sex, smoking history and NPC family history were significant between NPC cases and healthy controls (Table 1). There were no statistically significant differences in age, sex, smoking history and NPC family history between the early-stage and advanced-stage cases (Table 1).

The characteristics of the 320 cases and 320 healthy controls in the training stage are shown in Table 2. In this stage, the controls and NPC cases were matched by sex and age to prevent bias. Differences in age, smoking history and sex were not statistically significant, while differences in NPC family history were significant between the cases and controls.

**Table 1  Characteristics of the total population.**

| Categories | NPC cases (N = 554), n (%) | | | | Controls (N = 10,340) | |
|---|---|---|---|---|---|---|
| | Early-stage NPC cases (n = 73) | Advanced-stage NPC cases (n = 481) | Total | Pᵃ | n (%) | Pᵇ |
| Sex | | | | 0.356 | | <0.001 |
| Male | 49(67.12) | 348(72.35) | 397(71.66) | | 3,959(38.29) | |
| Female | 24(32.88) | 133(27.65) | 157(28.34) | | 6,381(61.71) | |
| Age (years) | | | | 0.546 | | <0.001 |
| 30~ | 7(9.60) | 26(5.41) | 33(5.96) | | 1,405(13.59) | |
| 35~ | 4(5.48) | 35(7.28) | 39(7.04) | | 1,439(13.92) | |
| 40~ | 6(8.22) | 64(13.31) | 70(12.64) | | 1,366(13.21) | |
| 45~ | 15(20.55) | 94(19.54) | 109(19.68) | | 1,512(14.62) | |
| 50~ | 14(19.18) | 86(17.88) | 100(18.05) | | 1,244(12.03) | |
| 55~ | 8(3.70) | 73(15.18) | 81(14.62) | | 997(9.64) | |
| 60~ | 11(15.07) | 71(14.76) | 82(14.80) | | 878(8.49) | |
| 65~69 | 8(10.96) | 32(6.65) | 40(7.22) | | 1,499(14.50) | |
| Smoking history | | | | 0.584 | | <0.001 |
| Yes | 24(32.88) | 174(36.17) | 198(35.74) | | 1,670(16.15) | |
| No | 49(67.12) | 307(63.83) | 356(64.26) | | 8,670(83.85) | |
| NPC family history | | | | 0.566 | | <0.001 |
| Yes | 6(8.22) | 50(10.4) | 56(10.11) | | 198(1.91) | |
| No | 67(91.78) | 431(89.60) | 498(89.89) | | 10,142(98.09) | |

**Notes.**

ᵃDifferences in sex, age, smoking history and NPC family history between early-stage and advanced-stage NPC cases were compared by the $\chi^2$ test.

ᵇDifferences in the baseline information distributions of the NPC cases and controls were compared by the $\chi^2$ test.

There were no statistically significant differences in sex, age, smoking history, or NPC family history between the early-stage and advanced-stage cases (Table 2).

In the verification stage, a total of 10,254 subjects were enrolled, including 234 NPC cases and 10,020 healthy controls. Of the 10,254 subjects, 3,897 (38.00%) were men, and the mean age was 48.57 ± 11.61 years. Among the 10,254 subjects, 1,664 (16.23%) had a smoking history, and 215 (2.10%) had a family history of NPC.

## Comparison of levels of markers in NPC patients and healthy controls

The rODs of VCA-IgA and EBNA1-IgA, PROB value and combination value in NPC patients and healthy controls were showed in Fig. 1. The $t$ tests showed that the means of markers in NPC patients were all significantly higher than those in healthy controls ($p < 0.001$).

## Comparison of levels of markers in early-stage and advanced-stage NPC patients

Of the 554 NPC patients, 73 (13.18%) were early-stage. The levels of VCA-IgA EBNA1-IgA, PROB and combination in early-stage and advanced-stage NPC patients were showed in Fig. 2. The differences in VCA-IgA, EBNA1-IgA, PROB and combination were not significant between early-stage and advanced-stage NPC patients by $t$ tests ($p > 0.05$).

**Table 2  Characteristics of the training stage population.**

| Categories | NPC cases (N = 320), n (%) | | | | Controls (N = 320) | |
|---|---|---|---|---|---|---|
| | Early- stage NPC cases (n = 40) | Advanced- stage NPC cases (n = 280) | Total | $P^a$ | n (%) | $P^b$ |
| Sex | | | | 0.377 | | 0.661 |
| Male | 26 (67.74) | 201 (74.89) | 227(70.94) | | 232 (72.50) | |
| Female | 14 (32.26) | 79 (25.11) | 93 (29.06) | | 88(27.50) | |
| Age (years) | | | | 0.419 | | 0.967 |
| 30~ | 3 (7.50) | 16 (5.71) | 19 (5.94) | | 23 (7.19) | |
| 35~ | 2 (5.00) | 23 (8.21) | 25 (7.81) | | 30 (9.38) | |
| 40~ | 1 (2.50) | 41 (14.64) | 42 (13.13) | | 40 (12.50) | |
| 45~ | 8 (20.00) | 55 (19.64) | 63 (19.69) | | 58 (18.13) | |
| 50~ | 7 (17.50) | 46 (16.43) | 53 (16.56) | | 48 (15.00) | |
| 55~ | 6 (15.00) | 42 (15.00) | 48 (15.00) | | 45 (14.06) | |
| 60~ | 9 (22.50) | 42 (15.00) | 51 (15.94) | | 53 (16.56) | |
| 65~69 | 4 (10.00) | 15 (5.36) | 19 (5.94) | | 23 (7.19) | |
| Smoking history | | | | 0.824 | | 0.235 |
| Yes | 13 (32.50) | 96 (34.29) | 109 (34.06) | | 95 (29.69) | |
| No | 27 (67.50) | 184 (65.71) | 211 (65.94) | | 225 (70.31) | |
| NPC family history | | | | 0.097 | | <0.001 |
| Yes | 1 (2.50) | 32 (11.42) | 33 (10.31) | | 6 (1.88) | |
| No | 39 (97.50) | 248(88.58) | 287 (89.69) | | 314 (98.12) | |

**Notes.**

[a] Differences in sex and smoking history between early-stage and advanced-stage NPC cases were compared by the $\chi^2$ test. Differences in age and NPC family history between early-stage and advanced-stage NPC cases were compared by Fisher's exact test.

[b] Differences in the baseline information distributions of the NPC cases and controls were compared by the $\chi^2$ test.

## Diagnostic value of the markers

The diagnostic performance of the markers is shown in Table 3 by using training samples. The combination achieved a sensitivity of 90.94% (95% CI [87.2%–93.8%]), a specificity of 92.50% (95% CI [89.0%–95.1%]) and an AUC of 0.978 (95% CI [0.969– 0.986]). The PROB achieved a sensitivity of 87.81% (95% CI [83.7%–91.2%]), a specificity of 95.94% (95% CI [93.2%–97.8%]) and an AUC of 0.972 (95% CI [0.962– 0.982]). The VCA-IgA had a sensitivity of 84.06% (95% CI [79.6%–87.9%]), a specificity of 91.25% (95% CI [87.6%–94.1%]) and an AUC of 0.947 (95% CI [0.932– 0.963]). The sensitivity, specificity and AUC of EBNA1-IgA were 87.81% (95% CI [83.7%–91.2%]), 85.00% (95% CI [80.6%–88.7%]), and 0.935 (95% CI [0.917–0.953]), respectively.

Compared to the AUCs of VCA-IgA ($p < 0.001$) ,EBNA1-IgA ( $p < 0.001$), and PROB ($p < 0.01$), the combination yielded a higher AUC (Table 3 and Fig. 3) by using training samples. The differences in the sensitivities of the markers between early-stage and advanced-stage NPC patients were not significant by using verification samples ($p > 0.05$, Table 4). Compared with each marker alone by McNemar test, the combination had a higher sensitivity for early-stage NPC patients (Table 4).

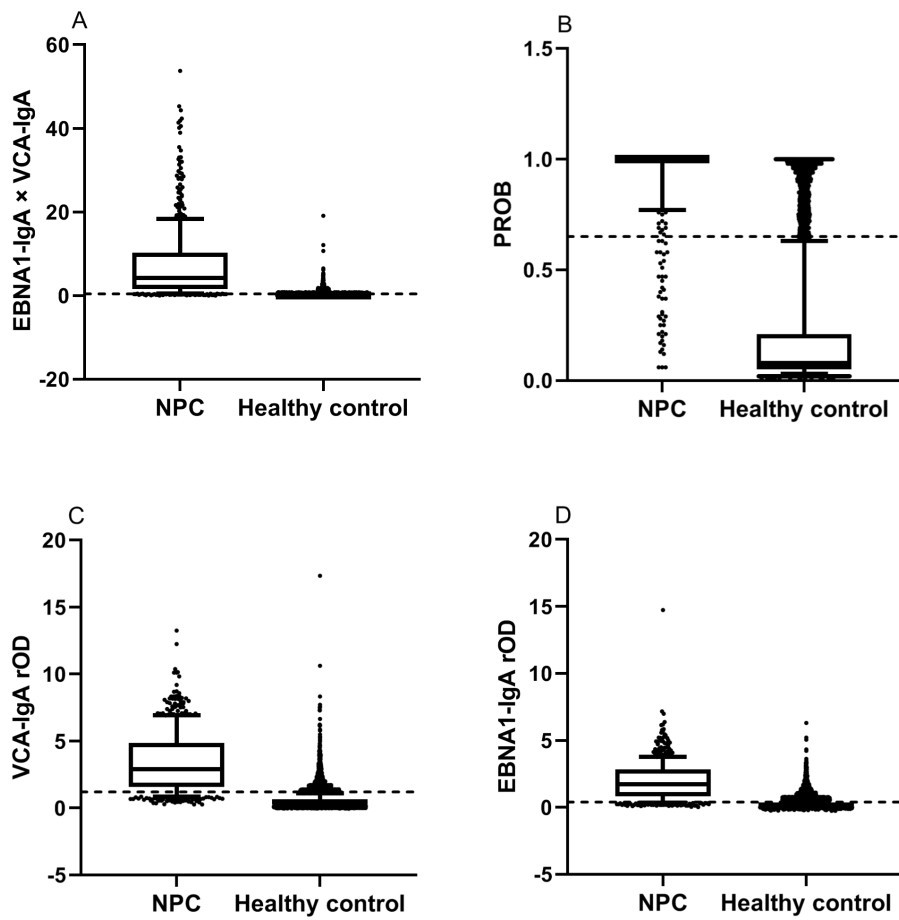

**Figure 1 Comparison of levels of markers in NPC patients and healthy controls by *t* tests.** The dotted lines represent cut-off values of the markers. Each box indicates 25/75 percentiles. Whisker caps represent 10/90 percentiles. (A) The level of the combination. The level of the combination for NPC patients was higher than for healthy controls by the t test (p<0.001). (B) The level of the PROB. The level of the PROB for NPC patients was higher than for healthy controls by the t test (p<0.001). (C) The level of the VCA-IgA. The level of the VCA-IgA for NPC patients was higher than for healthy controls by the t test (p<0.001). (D) The level of the EBNA1-IgA. The level of the EBNA1-IgA for NPC patients was higher than for healthy controls by the t test (p<0.001).

The differences in sensitivities of EBNA1-IgA, PROB and the combination between man and female NPC patients were not significant by using verification samples ($p > 0.05$), while the sensitivity of VCA-IgA in man NPC patients was higher than in female NPC patients ($p = 0.047$, Table 5). The sensitivity differences of the markers in different age, smoking history and NPC family history were not statistically significant by by using verification samples ($p > 0.05$, Tables 6–8).

## Verifying the effect of the combination on NPC screening

A total of 10,253 subjects were enrolled to verify the NPC screening effect, including 234 NPC cases and 10,020 healthy controls sourced from the screening field. In this stage, the combination achieved an overall sensitivity of 91.45% (214/234), a sensitivity for early-stage
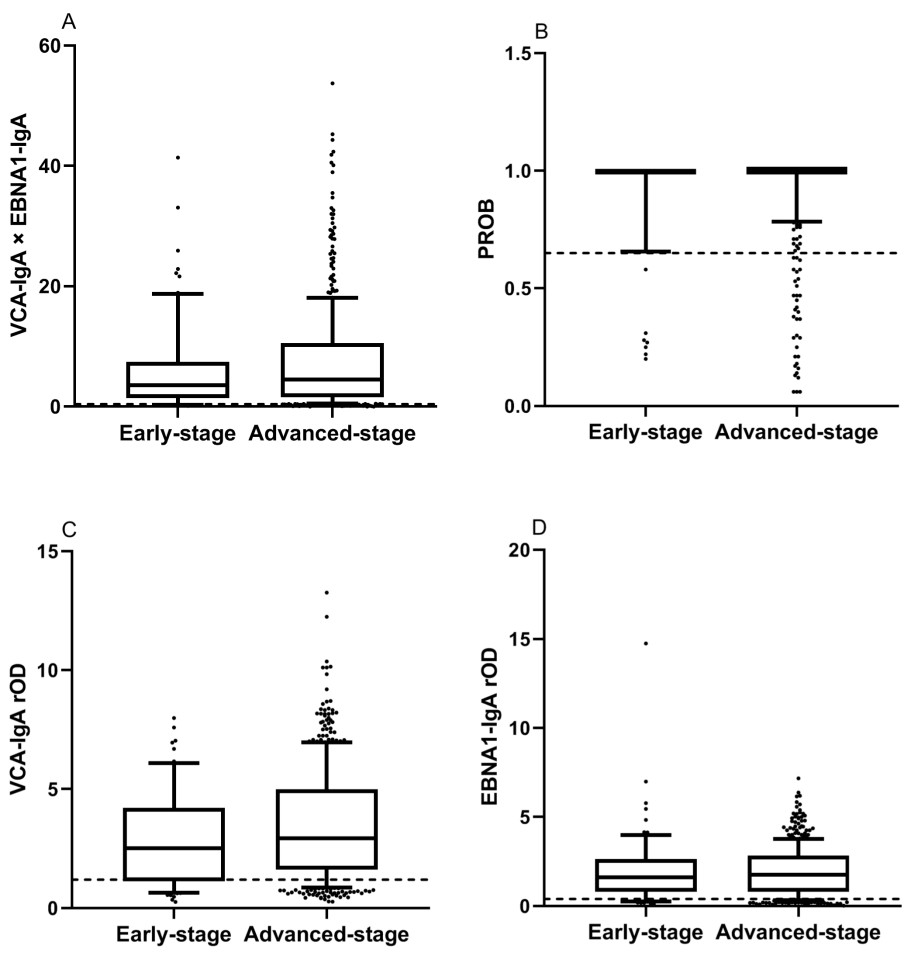

**Figure 2** **Comparison of levels of markers in early-stage and advanced-stage NPC patients by *t* tests.**
The dotted lines represent cut-off values of the markers. Each box indicates 25/75 percentiles. Whisker caps represent 10/90 percentiles. (A) The level of the combination. The difference in combination was not significant between early-stage and advanced-stage NPC patients by the t test (p>0.05). (B) The level of the PROB. The difference in PROB was not significant between early-stage and advanced-stage NPC patients by the t test (p>0.05). (C) The level of the VCA-IgA. The difference in VCA-IgA was not significant between early-stage and advanced-stage NPC patients by the t test (p>0.05). (D) The level of the EBNA1-IgA. The difference in EBNA1-IgA was not significant between early-stage and advanced-stage NPC patients by the t test (p>0.05).

NPC detection of 93.94% (31/33), a specificity of 93.45% (9364/10020), and an AUC of 0.978 (95% CI [0.968–0.987]).

# DISCUSSION

NPC is a main health problem that leads to a high health burden in southern China, especially in Guangdong province (*Cao, Simons & Qian, 2011*). EBV antibodies are widely applied in NPC screening. PROB calculated by logistic model were applied in NPC screening based on VCA-IgA and EBNA1-IgA (*Gao et al., 2017*; *Liu et al., 2013*; *Yu et al., 2018*). In this study, a combination calculated by multiplication model based on VCA-IgA

**Table 3** Diagnostic value of the markers.

| Marker | Cut-off value | Sensitivity (%) (95% CI) | Specificity (%) (95% CI) | AUC (95% CI) | P[a] |
|---|---|---|---|---|---|
| VCA-IgA × EBNA1-IgA | 0.429 | 90.94 (87.2, 93.8) | 92.50 (89.0, 95.1) | 0.978 (0.969, 0.986) | |
| PROB | 0.949 | 87.81 (83.7, 91.2) | 95.94 (93.2, 97.8) | 0.972 (0.962, 0.982) | <0.01 |
| VCA-IgA | 1.194 | 84.06 (79.6, 87.9) | 91.25 (87.6, 94.1) | 0.947 (0.932, 0.963) | <0.001 |
| EBNA1-IgA | 0.397 | 87.81 (83.7, 91.2) | 85.00 (80.6, 88.7) | 0.935 (0.917, 0.953) | <0.001 |

**Notes.**
[a] The AUC comparisons of the markers were performed by the $Z$ test.

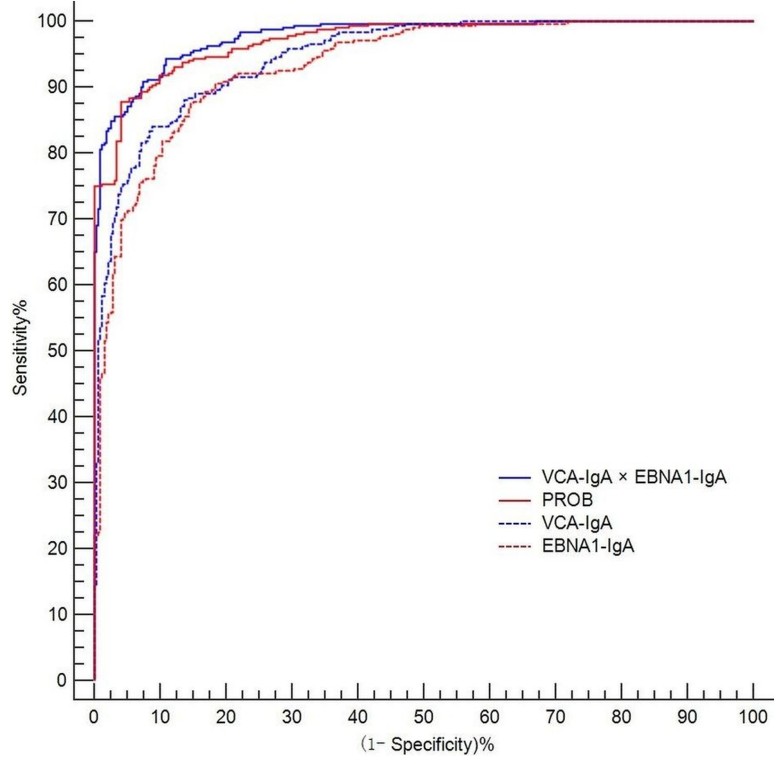

**Figure 3** **ROC curves for combination, PROB, VCA-IgA and EBNA1-IgA.** The axes are expressed as percentages.

and EBNA1-IgA was applied to NPC screening. The NPC screening effect of combination was evaluated and compared with the individual screening markers, PROB, VCA-IgA and EBNA1-IgA. Compared with PROB, VCA-IgA and EBNA1-IgA, the combination had a higher AUC of 0.978 (95% CI [0.969–0.986]). We found that the combination calculated by using a multiplication model can be applied to NPC screening.

In this study, a large number of healthy controls and 554 NPC patients were collected from the endemic areas, which is favourable for evaluating the performance of the markers for NPC screening. In the verification stage, 10,254 subjects were enrolled to verify the NPC screening effect. The combination achieved a sensitivity of 91.45%, a specificity of

**Table 4  Sensitivity differences for early-stage and advanced-stage NPC by using different markers.**

| Marker | Cut-off value | Sensitivity (%) | | $P^a$ |
|---|---|---|---|---|
| | | Early-stage NPC cases | Advanced-stage NPC cases | |
| VCA-IgA × EBNA1-IgA | 0.429 | 93.94 | 91.04 | 0.747 |
| PROB | 0.949 | 84.85[b] | 84.08 | 1.000 |
| VCA-IgA | 1.194 | 75.76[c] | 82.59 | 0.339 |
| EBNA1-IgA | 0.397 | 84.85[d] | 86.07 | 0.791 |

Notes.
[a]The sensitivity differences between early-stage and advanced-stage NPC were compared by Fisher's exact test
[b]Compared with combination, the PROB had a lower sensitivity for early-stage NPC patients by McNemar test ($p < 0.001$).
[c]Compared with combination, the VCA-IgA had a lower sensitivity for early-stage NPC patients by McNemar test ($p < 0.001$).
[d]Compared with combination, the EBNA1-IgA had a lower sensitivity for early-stage NPC patients by McNemar test ($p < 0.001$).

**Table 5  Sensitivity differences for man and female NPC by using different markers.**

| Marker | Cut-off value | Sensitivity (%) for NPC cases | | $P^a$ |
|---|---|---|---|---|
| | | Man | Female | |
| VCA-IgA$^x$ EBNA1-IgA | 0.429 | 91.76 | 90.63 | 0.781 |
| PROB | 0.949 | 85.29 | 81.25 | 0.450 |
| VCA-IgA | 1.194 | 84.71 | 73.44 | 0.047 |
| EBNA1-IgA | 0.397 | 85.88 | 85.93 | 0.991 |

Notes.
[a]The sensitivity differences between man and female NPC were compared by $\chi^2$ test.

**Table 6  Sensitivity differences for different ages of NPC patients by using different markers.**

| Marker | Sensitivity (%) for different ages (years) of NPC patients | | | | | | | | $P^a$ |
|---|---|---|---|---|---|---|---|---|---|
| | 30~ | 35~ | 40~ | 45~ | 50~ | 55~ | 60~ | 65~69 | |
| VCA-IgA × EBNA1-IgA | 100.00 | 85.71 | 89.29 | 91.30 | 85.11 | 100.00 | 93.55 | 90.48 | 0.274 |
| PROB | 85.71 | 78.57 | 78.57 | 84.78 | 83.00 | 90.91 | 87.10 | 80.95 | 0.904 |
| VCA-IgA | 78.57 | 78.57 | 85.71 | 80.43 | 78.72 | 84.85 | 87.10 | 76.19 | 0.950 |
| EBNA1-IgA | 92.86 | 78.57 | 85.71 | 86.96 | 85.11 | 93.94 | 83.87 | 76.19 | 0.674 |

Notes.
[a]The sensitivity differences in different ages of NPC patients were compared by Fisher's exact test.

93.45% and an AUC of 0.978 (95% CI [0.968–0.987]). These results demonstrated that the combination calculated by using a multiplication model was effective for NPC screening.

In this study, the levels of markers (PROB, VCA-IgA, EBNA1-IgA and combination) in NPC patients were higher than in healthy controls. It was consistent with the results of previous reports (*Liu et al., 2012*). We found the difference in sensitivities of the combination in different age, sex, smoking history and NPC family history were not statistically significant. The VCA-IgA had a higher sensitivity for man NPC patients than female NPC patients by using verification samples. Since the *P* value (0.047) was very close

**Table 7  Sensitivity differences for different smoking history NPC by using different markers.**

| Marker | Cut-off value | Sensitivity (%) for NPC | | $P^a$ |
|---|---|---|---|---|
| | | smoking | no smoking | |
| VCA-IgA$^x$ EBNA1-IgA | 0.429 | 91.01 | 91.72 | 0.850 |
| PROB | 0.949 | 87.64 | 82.07 | 0.257 |
| VCA-IgA | 1.194 | 85.39 | 79.31 | 0.243 |
| EBNA1-IgA | 0.397 | 86.52 | 85.52 | 0.831 |

Notes.
[a]The sensitivity differences for different smoking history NPC were compared by $\chi^2$ test.

**Table 8  Sensitivity differences for NPC with and without NPC family history by using different markers.**

| Marker | Cut-off value | Sensitivity (%) | | $P^a$ |
|---|---|---|---|---|
| | | NPC with NPC family history | NPC without NPC family history | |
| VCA-IgA$^\times$ | | | | |
| EBNA1-IgA | 0.429 | 91.30 | 91.47 | 1.000 |
| PROB | 0.949 | 78.26 | 84.83 | 0.412 |
| VCA-IgA | 1.194 | 86.96 | 81.04 | 0.487 |
| EBNA1-IgA | 0.397 | 73.91 | 87.02 | 0.082 |

Notes.
[a]The sensitivity differences between NPC with and without NPC family history were compared by $\chi^2$ test.

to 0.05, and the verification sample size was not very large. The difference in sensitivity of VCA-IgA between man and female NPC patients may be due to the random fluctuation.

In the present study, the AUCS, sensitivities and specificities of VCA-IgA and EBNA1-IgA were lower than those of the combination, showing that the combination was more effective in diagnosis. The AUC of the combination was a little bit larger than the PROB. There was a slight increase (3.13%) in the sensitivity of the combination compared to the sensitivity of the PROB. The specificity was lower for the combination (92.50%) than for the PROB (95.94%). In areas with high NPC incidence, the increased sensitivity means that more early-stage NPC patients will be detected and treated early, while the decreased specificity may lead to an increased false positive rate and increased costs of the screening program.

The present study had some limitations. First, there was some bias in identifying the 10340 subjects as healthy controls because not all healthy controls underwent an examination by fibreoptic endoscopic examination. Second, since the study population was obtained from provinces with a high risk of NPC, the results may be limited for application in other populations. Third, the sample size of the early-stage NPC patients was not large enough in this study. There was some bias in estimating sensitivity for early-stage NPC patients.

## CONCLUSIONS

We successfully developed a combination of two ELISA tests based on VCA-IgA and EBNA1-IgA to improve the effect of NPC screening by using a multiplication model. Compared with VCA-IgA and EBNA1-IgA individually, the combination had an improved diagnostic performance. The AUC and sensitivity of the combination were slightly higher than those of the PROB, while the specificity was lower for the combination than for the PROB. The results suggested that the combination was effective and can be an option for NPC screening.

## ACKNOWLEDGEMENTS

We thank the staff from Jiangmen Jianghai District People's Hospital (Chunlai Zhang and Yanshuang Lu), Waihai Public Health Service Center (Guozheng Zhou and Wenguang Lin), Jiangnan Public Health Service Center (Cannong Liang and Chunhua Xiong), Hetang Public Health Service Center (Wenwei Li), and all the other staff involved in this programme for their hard work.

### Funding

This work was supported by the Early Detection of Cancer Project in China (grant number: 2018-48). The funders had no role in study design, data collection and analysis, decision to publish, or preparation of the manuscript.

### Grant Disclosures

The following grant information was disclosed by the authors:
Early Detection of Cancer Project in China:  2018-48.

### Competing Interests

The authors declare there are no competing interests.

### Author Contributions

- Dongping Rao conceived and designed the experiments, analyzed the data, prepared figures and/or tables, authored or reviewed drafts of the paper, and approved the final draft.
- Meiqin Fu and Yanming Huang conceived and designed the experiments, performed the experiments, analyzed the data, prepared figures and/or tables, authored or reviewed drafts of the paper, and approved the final draft.
- Yingjie Chen, Xin Zhang, Zhongxiao Li and Yongxing Chen performed the experiments, prepared figures and/or tables, and approved the final draft.
- Qing Liu conceived and designed the experiments, prepared figures and/or tables, and approved the final draft.
- Lin Xiao, Haitao Li, Jieying Chen and Jin Hu performed the experiments, authored or reviewed drafts of the paper, and approved the final draft.

- Yongyi He analyzed the data, prepared figures and/or tables, and approved the final draft.

## Human Ethics

The following information was supplied relating to ethical approvals (i.e., approving body and any reference numbers):

This study was approved by the Clinical Research Ethics Committee of the Jiangmen Central Hospital (2019-28).

## Data Availability

The raw measurements are available in the Supplementary Files.

## Supplemental Information

Supplemental information for this article can be found online at http://dx.doi.org/10.7717/peerj.10254#supplemental-information.

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
