# Peer review of "A combination of two ELISA tests for nasopharyngeal carcinoma screening in endemic areas based on a case-control study"

_PeerJ, doi:10.7717/peerj.10254_

## Round 0.1 · original submission · Minor Revisions

Thank you for submitting your manuscript. You will note that our reviewers provided detailed feedback to improve your revision. I hope that you will consider resubmission.

·

Basic reporting

No comments

Experimental design

See comments below

Validity of the findings

See comments below

Additional comments

In this study, the authors evaluated the performance of EBV VCA and EBNA1 IgA antibody screening for NPC detection. Using data from over 550 NPC hospital-based cases and over 10,000 individuals who participated in an NPC screening program, the authors evaluate the performance characteristics of VCA IgA alone, EBNA1 IgA alone and a multiplicative combination of these 2 assay results for the identification of NPC. This is a large effort that explores a question of public health importance in regions with high incidence of NPC. Of note, performance of these same antibody tests has previously been evaluated/published using a combination risk score based on a logistic model, as appropriately noted by the authors.

The following suggestions/questions are offered to the authors:
1. It is unclear how many of the 550+ hospital cases included in this study derived from individuals who participated in the 10,000+ screening program. Assuming a subset of the 550+ cases were identified via screening, an additional analysis that restricts to individuals who participated in the screening study would be of interest (i.e., exclude NPC cases identified via self-referral of symptomatic individuals). Such an analysis would more accurately evaluate the performance of EBV-based antibody screening for the detection of NPC in the study population.
2. Since the same two assays have been previously used for NPC screening using a scoring system defined using a logistic model, it would be of interest to directly compare the performance of the published logistic model score to the present multiplicative score. Such an analysis would be highly informative to determine whether the multiplicative approach has improved performance compared to the already existent logistic model-based score.
3. It is unclear why a multiplicative score was chosen. The choice of such an approach should be further justified/explained. In addition, other methods of defining the VCA/EBNA1 combined risk score could be considered. At a minimum, an approach that considered the additive effect of the 2 assays could be considered.
4. While 70% of cases were male, <40% of controls were male. Given that rates of NPC differ considerably by sex, an analysis that evaluates the performance of the antibody-based score separately for men and women would be informative.

Reviewer 2 ·

Basic reporting

1) literature reference is needed for the multiplication model.

2) Figure 1 and Figure 2 should be converted to boxplot with marker cut-off value indicated in graphs.

Experimental design

1) Line 226 to 232 in manuscript - The description of results in these lines seems more appropriate to be under the Results sub-title "Diagnostic value of the markers" than under the Results sub-title "Diagnostic accuracies of the markers". For results that come under diagnostic accuracy, it is hoped that findings related to true positive (TP), false positive (FP), true negative (TN), false negative (FN), accuracy [formula = (TP + TN)/(TP + TN + FP + FN)] would be reported.

2) Diagnostic values of combined VCA/IgA and EBNA1/IgA markers had been reported by other case-control and prospective studies and shown to have higher diagnostic values than single markers (Gao et al., 2017; Liu et al., 2012; Yu et al., 2018). These studies utilized probability calculated by logistic regression while the authors of this manuscript applied multiplication model. This manuscript could provide a bit more justification about the usage of multiplication model and also perhaps conduct more analysis to cover the knowledge gap - compare the performance of these two modeling methods using same data set and discuss about their practicability and limitations.

Validity of the findings

1) In discussion, the authors mentioned that "The cost of NPC screening with EBV DNA test was expensive (Harris et al., 2019)...". However, this sentence is referencing to a report that made estimation from non-endemic area (Harris et al., 2019).

2) In discussion, the authors mentioned that "The diagnostic performance of EBV DNA test for early-stage NPC patients was not satisfactory (Ji et al., 2014)". The authors should include more recent studies in discussion (for examples, PMID: 28792880 and PMID: 31469434).

---

## Round 0.2 · Minor Revisions

Thank you for your work in addressing the reviewers' comments. Please see their follow-up comments to address some remaining minor revisions. Thank you!

·

Basic reporting

No new comment

Experimental design

No new comment

Validity of the findings

No new comment

Additional comments

I have reviewed the changes made to the manuscript and thank the authors for taking the time to respond to my previous comments. I have a couple remaining comments related to the new results summarized in Table 3.

1. Could the authors please confirm that the p-value comparing the AUC for the multiplicative (0.978; 95% CI = 0.969-0.986) and PROB (0.972; 95% CI = 0.962-0.982) models is <0.01? The 2 AUC estimates differ only in the 3rd decimal place so it is strange that the p-value reported is so small.

2. Regardless of whether or not the AUC for these 2 approaches is statistically significantly different, the difference is very minor and the authors should therefore comment on the practical implications of such a small change when deciding whether or not the multiplicative model truly improves performance over the currently used PROB model.

3. In responding to #2 above, please account for the fact that the sensitivity for the 2 approaches is very similar (91% and 88%) and that the specificity is lower for the multiplicative model (92%) than for the PROB model (96%). Since every 1% decrement in Specificity has a big impact on false(+) and programmatic costs, the fact that the PROB model has improved specificity over the newly evaluated multiplicative approach should be discussed. It is important for both the Abstract and Discussion to reflect the important finding that the PROB model is still likely the preferred way to utilize VCA/EBNA1 IgA antibody results for NPC screening and that the multiplicative model does not greatly improve on that approach overall and, in fact, has worst specificity and positive predictive value than the PROB approach.

Reviewer 2 ·

Basic reporting

1) Raw data of this study (peerj-48783-raw_data.xls):
column B title "NA1-IgA rOD" should be "EBNA1-IgA rOD"
list "PROB" values for each individual

2) change whiskers of box plots in Figure 1 and 2 to indicate the 10th percentile and 90th percentile and describe this in figure legend.

3) when describing the serological results, be consistent in using "/" or "-" (e.g. VCA-IgA or VCA/IgA)

4) Sensitivity and specificity values are stated using percentage in Table 3 but mentioned as fractions in manuscript (line 227 to 235).

5) are Figure 3, Table 3, Table 4 and Table 5 showing results from all or just samples the training stage? please state this in the manuscript

Experimental design

no comment

Validity of the findings

no comment

Additional comments

1) Serological results for male NPC patients and female NPC patients are not shown but VCA-IgA was reported as having higher sensitivity in detecting male NPC patients as compared to female NPC patients (Table 5). There should be some explanation or speculation about this difference in Discussion.

2) Serological results may or may not be affected by other factors stated in Table 1 (age, NPC family history and smoking history). The authors should analyse and report the findings.

3) As compared to the logistic regression model ("PROB"), the multiplication model ("combination") developed in this study had small decrease in specificity to detect all NPC (Table 3), but slight increase in sensitivity to detect all NPC (Table 3) and remarkable improvement in sensitivity to detect early stage NPC (Table 5). Are these results consistent in the training stage as well as in the verification stage? If so, what are the implications?

4) The results of verification stage were mentioned twice in separate paragraphs within Discussion.

---

## Round 0.3 · accepted · Accept

I am pleased to notify you that your paper has been accepted. Please note the minor suggested edits to improve readability.

Reviewer 2 ·

Basic reporting

The authors mentioned that "The verification sample data set was used to make Table 4 ,Table 5,Table 6,Table 7 and Table 8". Therefore in the manuscript, line 243 starting from "The differences in the sensitivities of the markers between early-stage and advanced-stage NPC patients were not significant by using verification samples(p>0.05, Table 4)..." to the end of line 255 should be moved to line 263 under the Results sub-topic "Verifying the effect of the combination on NPC screening".

Experimental design

no comment

Validity of the findings

no comment